# Sn(IV)-free tin perovskite films realized by in situ Sn(0) nanoparticle treatment of the precursor solution

Tomoya Nakamura [1], Shinya Yakumaru [1], Minh Anh Truong[1], Kyusun Kim[1], Jiewei Liu[1], Shuaifeng Hu [1], Kento Otsuka[1], Ruito Hashimoto[1], Richard Murdey[1], Takahiro Sasamori [2], Hyung Do Kim[3], Hideo Ohkita [3], Taketo Handa[1], Yoshihiko Kanemitsu [1] & Atsushi Wakamiya [1✉]

The toxicity of lead perovskite hampers the commercialization of perovskite-based photovoltaics. While tin perovskite is a promising alternative, the facile oxidation of tin(II) to tin(IV) causes a high density of defects, resulting in lower solar cell efficiencies. Here, we show that tin(0) nanoparticles in the precursor solution can scavenge tin(IV) impurities, and demonstrate that this treatment leads to effectively tin(IV)-free perovskite films with strong photoluminescence and prolonged decay lifetimes. These nanoparticles are generated by the selective reaction of a dihydropyrazine derivative with the tin(II) fluoride additive already present in the precursor solution. Using this nanoparticle treatment, the power conversion efficiency of tin-based solar cells reaches 11.5%, with an open-circuit voltage of 0.76 V. Our nanoparticle treatment is a simple and broadly effective method that improves the purity and electrical performance of tin perovskite films.

[1] Institute for Chemical Research, Kyoto University, Gokasho, Uji, Kyoto 611-0011, Japan. [2] Graduate School of Natural Sciences, Nagoya City University, Yamanohata 1, Mizuho-cho, Mizuho-ku, Nagoya, Aichi 467-8501, Japan. [3] Department of Polymer Chemistry, Graduate School of Engineering, Kyoto University, Katsura, Nishikyo-ku, Kyoto 615-8510, Japan. ✉email: wakamiya@scl.kyoto-u.ac.jp

Organic–inorganic hybrid halide perovskites are promising optoelectronic materials with a wide range of applications such as solar cells, light-emitting diodes, and photonic devices[1]. The use of toxic lead-based perovskite materials in these devices is, however, a significant impediment to commercialization. Replacing lead (Pb) with environmentally friendly elements, such as tin (Sn)[2], germanium (Ge)[3], copper (Cu)[4], antimony (Sb)[5], or bismuth (Bi)[6], is therefore essential. Among Pb-free perovskite candidates, Sn-based perovskites have favorable optoelectronic properties[7,8], and are currently considered as the most promising candidate for the development of Pb-free perovskite solar cells (PSCs). The reported power conversion efficiencies (PCEs) of Sn-based PSCs are still around 10%, significantly lower than those of Pb-based cells which presently exceed 25%[9]. Challenges for Sn-based perovskites compared with their Pb counterparts include (i) poor film morphology due to the rapid crystallization, (ii) disadvantageous energy level alignment with common charge transport layers, and (iii) facile oxidation of divalent Sn(II) into Sn(IV) which leads to the increased nonradiative charge recombination in the perovskite films[10–12]. Combined, these characteristics tend to result in poor device performance generally, and low open-circuit voltages ($V_{OC}$) in particular.

Several strategies have been pursued to improve the performance of Sn-based PSCs, including solvent engineering[13–15], crystal growth control[16], and compositional variation[17–19]. Various reductants including hypophosphorous acid[20], hydrazine vapor[21], Sn bulk powder[22], and hydroquinone sulfonic acid[23] were found to be moderately effective in decreasing the concentration of Sn(IV). SnF$_2$ additive in the precursor solution was also found to be crucial to prevent the heavy $p$-doping of the perovskite films[24–26]. A PCE of 10.2% with a $V_{OC}$ of 0.6 V was achieved by passivating the perovskite surface with ethylene diamine (EDA)[27]. This was only very recently surpassed by Ning et al., by combining two-dimensional perovskite structure with a shallow-LUMO electron transport layer[28].

We have previously developed thoroughly purified starting materials for Sn-based PSCs[29]. Although this resulted in improved device performance, the perovskite layers still contain Sn(IV) impurities which formed after the starting materials were purified, most likely by reactions with trace amounts of oxygen in the environment. If the Sn(IV) impurities are completely removed, the doping effect of Sn(IV) and/or any resultant trap states would be eliminated and the electrical properties of the perovskite layers would be further enhanced.

In the present work, we show how Sn(0) nanoparticles formed in situ in the precursor solution can scavenge these residual Sn(IV) impurities. The nanoparticles are formed by the reduction of SnF$_2$ in the precursor solution to Sn(0) by 1,4-bis(trimethylsilyl)-2,3,5,6-tetramethyl-1,4-dihydropyrazine (TM-DHP). This compound, known as 'Mashima reagent', is reported to have an unusually high reactivity, deriving from its $8\pi$ electron character[30–33]. Perovskite films fabricated using the purified precursor solution are verified by X-ray photoelectron spectroscopy (XPS) to be essentially free of Sn(IV) species. The efficient Sn(IV) scavenging is made possible by the high chemical selectivity of TM-DHP for SnF$_2$ over SnI$_2$. The Sn(IV)-free perovskite films show enhanced photoluminescence with elongated lifetimes. The effectiveness of the Sn(IV)-free tin perovskite films in electronic devices is demonstrated by fabricating Sn-based solar cells, which reach a PCE of 11.5% with a $V_{OC}$ of 0.76 V.

## Results

**Purity of SnI$_2$ reagents**. The purity of the SnI$_2$ starting material influences the physical properties of the tin perovskite. SnI$_2$ obtained from three different companies, Kojundo Chemical,

Sigma-Aldrich, and TCI, were compared with SnI$_2$(dmf)[29], our purified SnI$_2$ starting material complexed with dimethylformamide, in a mixed-cation FA$_{0.75}$MA$_{0.25}$SnI$_3$ perovskite composition (Supplementary Fig. 1). This particular perovskite formula was selected for its optimal charge extraction when paired with common hole transport materials such as PEDOT:PSS[15,34]. Following the widely used method for tin perovskite film formation, 10 mol% SnF$_2$ was also added to the precursor solution containing SnI$_2$, FAI, and MAI. SnF$_2$ is an additive which is important for the film growth control. The perovskite films were fabricated with our hot antisolvent treatment (HAT), which improves the film coverage[15]. No difference in the X-ray diffraction (XRD) patterns was observed for the films prepared from different SnI$_2$ sources (Supplementary Fig. 2a). In the optical absorption spectra, the film prepared from SnI$_2$ containing ca. 10 wt% SnI$_4$[29] can be differentiated by a blue-shifted absorption with a broader onset, while the other three samples were essentially identical (Supplementary Fig. 2b). The SnI$_2$ sources were, however, clearly differentiated through their photoluminescence properties. While the film prepared from our SnI$_2$(dmf) complex gave a longer PL lifetime and a higher intensity than the other samples (Supplementary Fig. 2c, d), the PL lifetimes (2–4 ns) are still short. Residual Sn(IV) is therefore most likely still present in the films.

**Sn(IV) content in perovskite films**. The issue of Sn(IV) impurities in the Sn perovskite film can be addressed by removing them from the perovskite precursor solution (Fig. 1). We envisioned that this would be accomplished by adding TM-DHP reductant. While we originally thought that TM-DHP would act as a scavenger by reducing Sn(IV) directly, the scavenger species is instead found to be Sn(0) nanoparticles created in the precursor solution by an in situ selective reaction of TM-DHP with SnF$_2$.

TM-DHP was synthesized as colorless crystals following the reported method (Supplementary Note 1)[30,31] and was characterized by $^1$H NMR, cyclic voltammetry ($E_{ox,1/2} = -0.57$ V vs. Fc/Fc$^+$, Supplementary Fig. 3), and single crystal X-ray structure analysis (Supplementary Note 2). When 1 mol% of TM-DHP was added to the perovskite precursor solution, the color immediately changed from clear yellow to yellowish-gray, turning back to clear yellow after stirring at 45 °C for ca. 15 min (Supplementary Fig. 4). The solution was filtered prior to film fabrication. The PL intensity increased and the lifetimes of the treated films also increased from 4.0 to 14.3 ns (Fig. 2a, b). When the amount of TM-DHP was increased to 5 mol% and 10 mol% compared with 10 mol% of SnF$_2$, the lifetime decreased to 6.6 and 0.8 ns, respectively. Considering that SnF$_2$ is consumed by reaction with TM-DHP, as discussed later, the correlation of PL lifetime with the amount of SnF$_2$ remaining in the precursor suggests that around 10 mol% SnF$_2$ is still needed as a moderator for optimal performance irrespective of the TM-DHP treatment. The PL peak position as well as the film morphology (Supplementary Fig. 5) were almost unchanged, confirming that neither TM-DHP nor any of its reaction products are incorporated into the perovskite structure to any significant extent. The scavenging ability of some alternative reductants, 1,4-bis(trimethylsilyl)-1,4-dihydropyrazine (DHP), GeCl$_2$·dioxane, Et$_3$SiH, Sn bulk powder[22], and tetramethylpyrazine (TMP)[16] were also investigated but none of the compounds trialed were found to be as effective as TM-DHP (Supplementary Fig. 6).

The amount of Sn(IV) in the perovskite films was estimated using XPS. From the Sn $3d^{5/2}$ spectra of 190 nm thick films (Supplementary Fig. 7), the addition of 1 mol% TM-DHP greatly reduced the content of Sn(IV) at the surface of perovskite films, from 15.5% to 5.3% (Fig. 2c). After argon etching to expose the

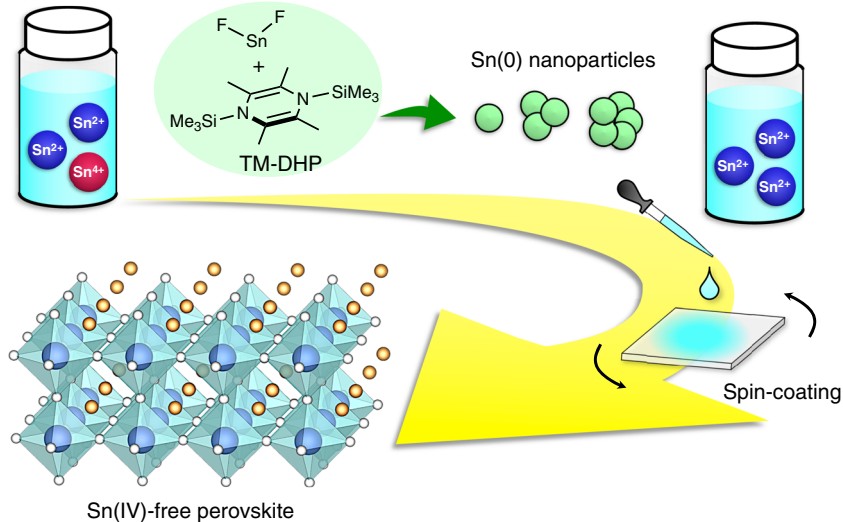

**Fig. 1 Schematic illustration of the Sn(IV) scavenging method.** 1,4-Bis(trimethylsilyl)-2,3,5,6-tetramethyl-1,4-dihydropyrazine (TM-DHP) is added to the tin perovskite precursor solution. Sn(0) nanoparticles formed by the reduction of $SnF_2$ by TM-DHP scavenge residual Sn(IV) impurities.

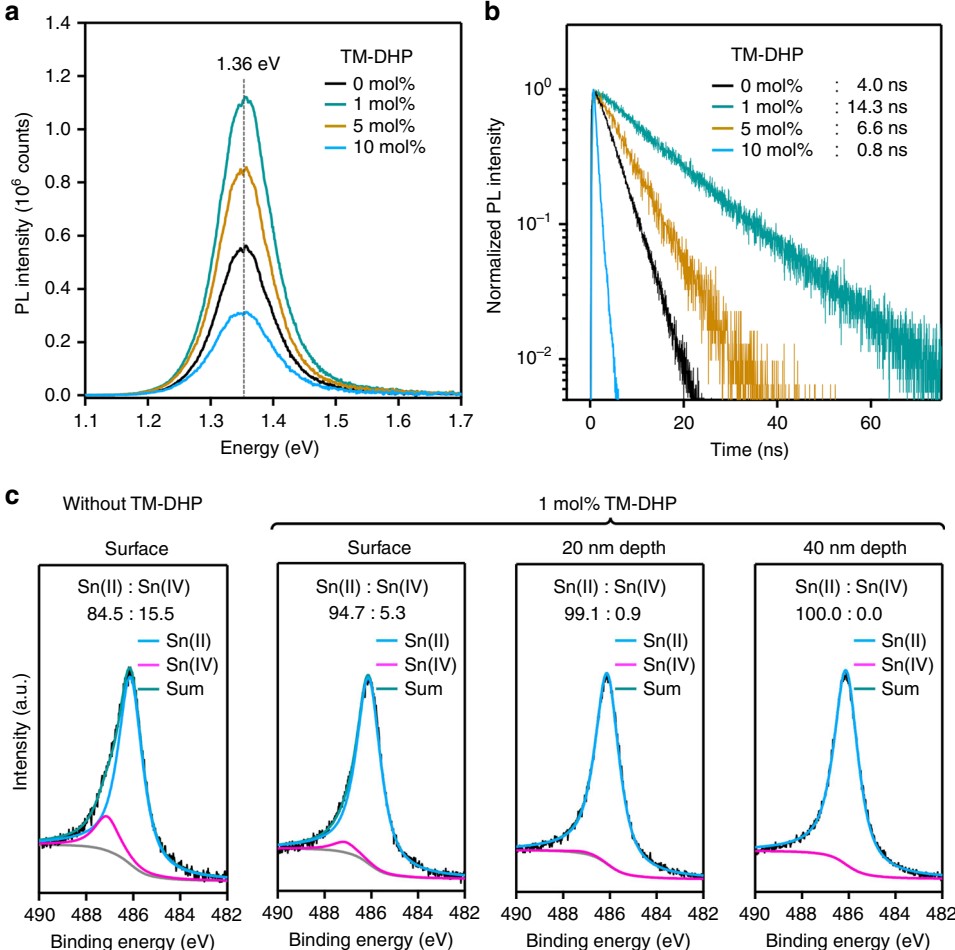

**Fig. 2 Effect of Sn(IV) scavenging by Sn nanoparticles. a** PL spectra and **b** PL decay curves of $FA_{0.75}MA_{0.25}SnI_3$ perovskite films prepared with 0, 1, 5, and 10 mol% of TM-DHP and 10 mol% $SnF_2$, excited at 688 nm with excitation fluence of 127 nJ cm$^{-2}$. **c** XPS Sn 3d$^{5/2}$ spectra of perovskite films.

bulk perovskite, the Sn(IV) content decreased to 0.9% (10 s etching, ca. 20 nm depth), and finally 0.0% (20 s etching, ca. 40 nm depth). Similar results were also obtained from fitting the Sn 3d$^{3/2}$ peak (Supplementary Fig. 8). We can therefore infer that the bulk perovskite is essentially free of Sn(IV) species. This

represents a significant advance compared with previous reports, where the amount of Sn(IV) in the bulk was over 5%[16,18,23,27]. The effect of varying the amount of $SnF_2$ was also investigated while keeping TM-DHP at 1 mol%. The bulk perovskite remained Sn(IV)-free even when the amount of $SnF_2$ was reduced

to 1 mol% (Supplementary Fig. 9). The Sn(IV) content at the perovskite surface decreased when the $SnF_2$ amount was increased, however, showing the positive effect of excess $SnF_2$ to suppress surface oxidation occurring after film fabrication. Optimal results were obtained with the ratio of 10 mol% $SnF_2$ and 1 mol% TM-DHP.

**Sn(IV) scavenging mechanism.** In order to explore the scavenging mechanism, [119]Sn NMR spectra were taken for various model systems. When 1 equivalent of TM-DHP was added to a solution of $SnI_4$ in DMSO-$d_6$, a very small amount of $SnI_2$ was formed while the majority of the starting $SnI_4$ was unreacted (Fig. 3a). On the addition of TM-DHP to a mixture of $SnI_4$ and $SnF_2$ (1:1), however, the peaks from $SnI_4$ disappeared and only the peak of $SnI_2$ was observed (Fig. 3b). [1]H NMR revealed that this reaction is accompanied by the formation of tetra-methylpyridine (TMP) and $Me_3SiF$ (Supplementary Fig. 10). Clearly, the reaction of $SnI_4$ with TM-DHP is greatly accelerated by the presence of $SnF_2$. When only TM-DHP and $SnF_2$ were mixed together, the peak from $SnF_2$ disappeared completely (Supplementary Fig. 11a). A gray suspension immediately formed after the addition of TM-DHP (Fig. 3c), which was confirmed by transmission electron microscopy (TEM) to contain nanoparticles up to 15 nm (Fig. 3d). The reduction of $SnI_4$ by TM-DHP is therefore most likely a two-step process where $SnF_2$ is selectively reduced by TM-DHP to form Sn(0) nanoparticles, followed by the reduction of the $SnI_4$ at the nanoparticle surface to form $SnI_2$ (Supplementary Fig. 12). The nanoparticles gradually aggregate in the precursor solution to form larger particles. The nanoparticles grow to ca. 60 nm after 30 s. Bulk metal precipitate was observed after stirring for 15 min (Supplementary Fig. 13). The solution was stirred for another 15 min, after which the metal precipitate was conveniently removed by filtration through a 0.45 μm PTFE filter. The formation of Sn(0) metal was confirmed by XPS measurement (Supplementary Fig. 14). $SnI_2$, meanwhile, was not reduced by TM-DHP (Supplementary Fig. 11b).

The selective reduction of $SnF_2$ against $SnBr_2$ and $SnCl_2$ was also confirmed by [119]Sn NMR (Supplementary Fig. 11c, d). This selectivity for $SnF_2$ is most likely derived from a strong affinity of the trimethylsilyl group in TM-DHP for fluoride. A plausible reaction mechanism for the formation of Sn(0) nanoparticles is given in Supplementary Fig. 15[32,33]. It should be noted that our Sn(IV) scavenging method using TM-DHP can be applied to various compositions of Sn-based perovskite optoelectronic materials (e.g. $ASnX_3$, X = I, Br, Cl), the only condition is the requirement that a small amount of $SnF_2$ be present in the precursor solution.

**Solar cell device characterization.** In order to demonstrate the effect of the Sn(IV) scavenging on the device performance, solar cell devices were fabricated with the following structure: ITO/PEDOT: PSS/$FA_{0.75}MA_{0.25}SnI_3$/$C_{60}$/bathocuproine (BCP)/Ag (Supplementary Fig. 16). Treating the precursor solution with 1 mol% TM-DHP caused the average PCE to increase from 6.6 to 8.9% (Fig. 4, Table 1). In addition to the increase in short-circuit current density ($J_{SC}$), the increase in $V_{OC}$ and fill factor (FF) confirms that charge carrier recombination is effectively suppressed. This result is consistent with the elongated PL lifetimes noted for the perovskite films. The PCE decreased, however, when the concentration of TM-DHP in the precursor solution was increased to 5 mol% or 10 mol%, again consistent with the PL lifetime data. The best result obtained for a device with 1 mol% TM-DHP was PCE = 9.9% (forward scan), $J_{SC}$ = 21.1 mA cm$^{-2}$, $V_{OC}$ = 0.63 V, and FF = 0.74 (Fig. 4a), with a small hysteresis (Supplementary Fig. 17a). The integrated $J_{SC}$ from the external quantum efficiency (EQE) was 21.2 mA cm$^{-2}$, comparable to the values obtained from the $J$–$V$ curve scan (Supplementary Fig. 17b). Stable power output for 600 s under AM1.5 G operation was confirmed (Supplementary Fig. 18). Evaluation of 40 independent cells showed high reproducibility with PCEs in a range from 8 to 10% (Fig. 4b, Supplementary Fig. 19). Increasing the thickness of the perovskite layer to 270 or 320 nm did not result in any increase in $J_{SC}$, suggesting that the charge carrier diffusion lengths do not exceed 200 nm (Supplementary Fig. 20).

The maximum $V_{OC}$ (0.63 V) is still low considering the optical band gap (1.36 eV for $FA_{0.75}MA_{0.25}SnI_3$). To clarify the origin of the large voltage loss of 0.73 V, the light intensity and temperature dependence of $V_{OC}$ were measured. While the estimated diode ideality factor was close to unity ($n_{id}$ = 1.1), the effective band gap was extrapolated to 0.74 eV, which is much smaller than the optical band gap (Supplementary Fig. 21). We can infer from this that the voltage loss is mainly due to the recombination at the charge extraction interfaces, rather than occurring within the perovskite layer. To address this, the perovskite surface was first treated with EDA[27], which increased the $V_{OC}$ up to 0.68 V (Fig. 5a, Table 2). Furthermore, a thin layer of $PC_{61}BM$ (<5 nm) was inserted to promote Ohmic contacts by minimizing the energy gap between the conduction band of perovskite and the LUMO energy level of the electron transport

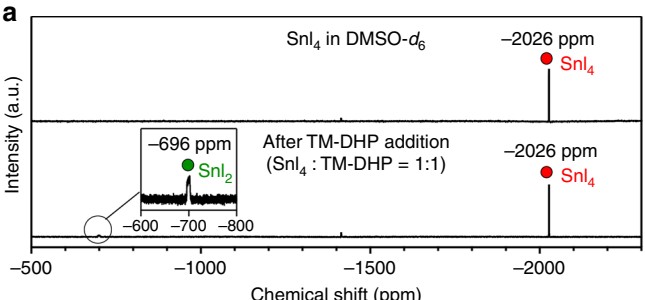

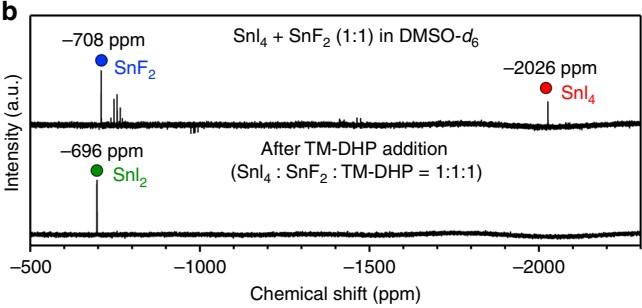

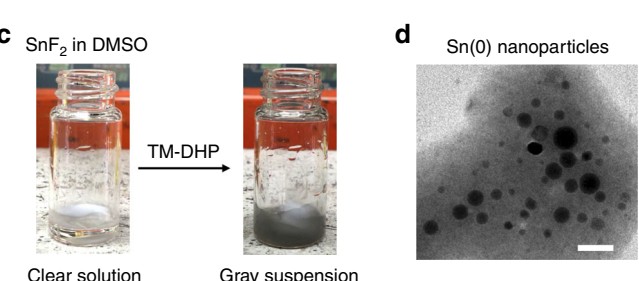

**Fig. 3 Elucidation of the Sn(IV) scavenging mechanism.** [119]Sn NMR spectra of **a** $SnI_4$, and **b** $SnI_4$ with $SnF_2$ in DMSO-$d_6$, before (top) and after (bottom) adding 1 equivalent of TM-DHP. **c** Photos of the $SnF_2$ solution before (left) and after (right) the addition of TM-DHP and **d** TEM image of the formed Sn nanoparticles. The scale bar is 20 nm.

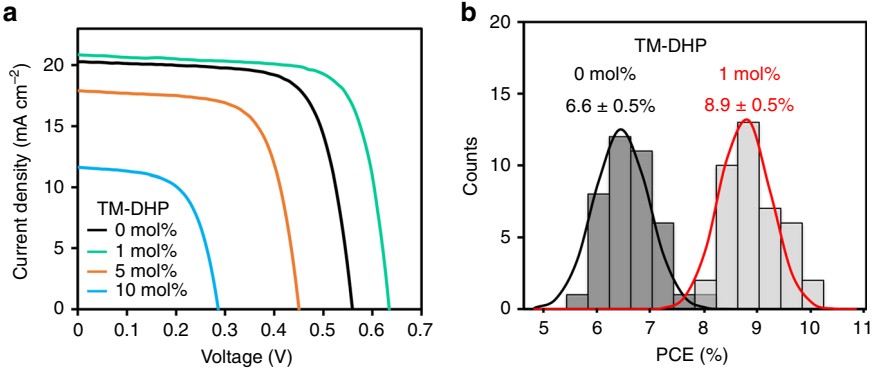

**Fig. 4 Performance of the Sn-based perovskite solar cell devices. a** *J–V* curves of Sn-based PSCs with device area of 0.1 cm², prepared by adding 0, 1, 5, and 10 mol% of reductant TM-DHP, respectively, under AM1.5 G, 100 mW cm⁻² irradiation. **b** Statistics of the PCE distribution of 40 cells with and without 1 mol% of TM-DHP.

**Table 1 Photovoltaic parameters of Sn-based PSCs with 0, 1, 5, and 10 mol% of TM-DHP.**

| TM-DHP (mol%) | $J_{SC}$ (mA cm⁻²) | $V_{OC}$ (V) | FF | PCE (%) |
|---|---|---|---|---|
| 0 | 20.3 (19.3 ± 0.7) | 0.56 (0.53 ± 0.02) | 0.72 (0.64 ± 0.04) | 8.1 (6.6 ± 0.5) |
| 1 | 21.1 (21.6 ± 0.5) | 0.63 (0.58 ± 0.03) | 0.74 (0.72 ± 0.02) | 9.9 (8.9 ± 0.5) |
| 5 | 18.2 (17.8 ± 0.4) | 0.45 (0.45 ± 0.01) | 0.68 (0.68 ± 0.02) | 5.6 (5.4 ± 0.2) |
| 10 | 11.6 (10.9 ± 0.9) | 0.29 (0.28 ± 0.01) | 0.61 (0.59 ± 0.03) | 2.0 (1.8 ± 0.3) |

Average values for 40 cells given in parenthesis.

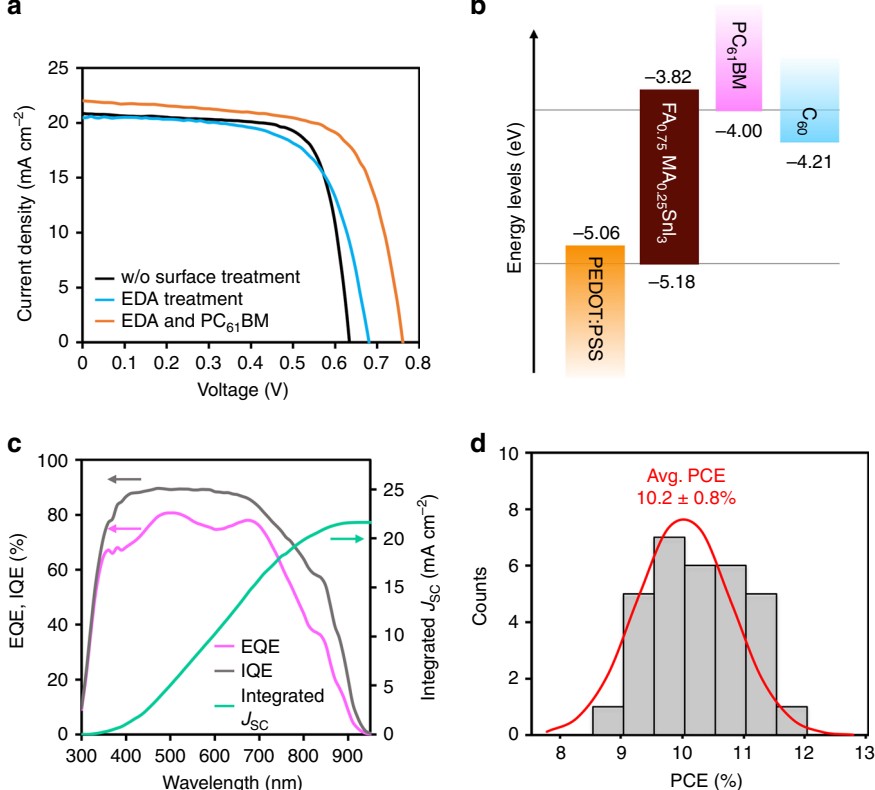

**Fig. 5 Solar cell device characterization after surface treatment. a** *J–V* curves of Sn-based PSCs with or without surface treatment with EDA and PC₆₁BM. **b** Energy levels diagram of the cell components. **c** EQE and IQE spectra of the best device with EDA and PC₆₁BM treatment. **d** Statistics of the PCE distribution of 30 cells.

**Table 2 Photovoltaic parameters of surface-treated Sn-based PSCs.**

| Surface treatment | $J_{SC}$ (mA cm$^{-2}$) | $V_{OC}$ (V) | FF | PCE (%) |
|---|---|---|---|---|
| EDA | 20.5 | 0.68 | 0.66 | 9.2 |
| EDA & PC$_{61}$BM | 22.0 (21.5 ± 0.9) | 0.76 (0.69 ± 0.05) | 0.69 (0.70 ± 0.02) | 11.5 (10.2 ± 0.8) |

Average values for 30 cells given in parenthesis.

layer (Fig. 5b and Supplementary Fig. 22), and to reduce non-radiative recombination at the interface. With this change, the PCE was improved to 11.5%, with $J_{SC} = 22.0$ mA cm$^{-2}$, $V_{OC} = 0.76$ V, and FF $= 0.69$ (Fig. 5a and Supplementary Fig. 23). The integrated $J_{SC}$ from EQE was 21.6 mA cm$^{-2}$ (Fig. 5c). A C$_{60}$ layer was necessary in our device structure, as confirmed by the much lower performance in its absence (Supplementary Fig. 24). High reproducibility was confirmed for 30 cells, with the average PCE of 10.2 ± 0.8% (Fig. 5d). The device also exhibited a high stability when stored under inert condition without encapsulation, showing no decrease in the performance after 50 days (Supplementary Fig. 25). An encapsulated device was tested by a professional institution, giving a certified PCE of 11.2% (Supplementary Fig. 26).

## Discussion
We have shown that Sn(IV) impurities in FA$_{0.75}$MA$_{0.25}$SnI$_3$ perovskite can be scavenged by Sn(0) nanoparticles formed in situ in the precursor solution by the selective reaction of TM-DHP with SnF$_2$. Perovskite layers fabricated with the purified precursor solution were verified by XPS to be essentially free of Sn(IV) ions. The Sn(IV)-free perovskite films exhibited strong photoluminescence with prolonged lifetimes and improved solar cell performance. Efficient scavenging is enabled by the highly selective reactivity of TM-DHP for SnF$_2$ over SnX$_2$ (X = I, Br, Cl), which likely stems from the strong affinity between the trimethylsilyl groups and the fluoride. In combination with interface modification by EDA and PC$_{61}$BM, devices with a $V_{OC}$ of up to 0.76 V and a PCE of 11.5% were achieved. Our nanoparticle-based scavenging method for Sn(IV) would be applicable to various kinds of Sn-based perovskites, not only for solar cells but also for light-emitting diodes and other optoelectronic devices. These investigations are currently in progress in our laboratory, and results will be reported in due course.

## Methods
**Materials**. Methylammonium iodide (MAI), formamidinium iodide (FAI), BCP, and SnI$_2$ (sublimed, 97%) were purchased from Tokyo Chemical Industry Co., Ltd. (TCI). SnF$_2$ (99%), SnI$_2$ (99.99%, trace metal basis) and ethylene diamine (EDA, 99.5%) were purchased from Sigma–Aldrich Co., Ltd. (Sigma-Aldrich). SnI$_2$ (99.9%, trace metal basis) was purchased from Kojundo Chemical Laboratory Co., Ltd (Kojundo Chemical). PEDOT:PSS (Clevious P VP AI 4083) was purchased from Heraeus Co., Ltd. C$_{60}$ (sublimed, 99.99%) was purchased from ATR Company. [6,6]-phenyl-C$_{61}$-butyric acid methyl ester (PC$_{61}$BM, 99.5%) was purchased from Solenne BV. MAI and FAI were recrystalized from ethanol and diethyl ether before use. SnI$_2$(dmf) complex was synthesized following the reported method[29]. EDA was distilled from CaH$_2$. Other materials were used as received. Dehydrated dimethylsulfoxide (DMSO, super dehydrated) was purchased from FUJIFILM Wako Pure Chemical Co., Ltd. Dimethylformamide (DMF) and chlorobenzene were purchased from Kanto Chemical. Co., Inc. All of these solvents were degassed by Ar gas bubbling for 1 h and further dried over molecular sieves in an Ar-filled glove box (O$_2$ < 0.1 ppm) before use. Tesa tape (tesa$^{®}$ 61562) was gifted from tesa tape K.K. Barrier film (PT7/25GT3) was gifted from OIKE & Co., Ltd.

**Perovskite layer fabrication**. The perovskite film preparation was conducted in an Ar-filled glove box (O$_2$ < 0.1 ppm). The FA$_{0.75}$MA$_{0.25}$SnI$_3$ perovskite precursor solution was prepared by mixing SnI$_2$ (335.3 mg, 0.90 mmol) or SnI$_2$(dmf) complex (401.0 mg, 0.90 mmol), FAI (116.1 mg, 0.68 mmol), MAI (35.8 mg, 0.23 mmol),

and SnF$_2$ (14.1 mg, 0.09 mmol, 10 mol%) in 1.0 mL DMSO to reach a concentration of 0.9 M. After stirring the solution at 45 °C for 30 min, a solution of reductant TM-DHP in DMF (0.9 M, 10 μL, 0.009 mmol) was added to reach the amount of 1.0 mol% for SnI$_2$. The color of precursor solution changed from clear yellow to yellowish-gray, and turned back to clear yellow after stirring the solution at 45 °C for ca. 15 min. After stirring for another 15 min, the solution was filtered through a 0.45 μm PTFE filter. Two hundred microliters of the precursor was used for spin coating, with a program set as 5 s acceleration to 5000 rpm, spinning at 5000 rpm for 60 s, and finally 1 s deceleration to stop. At 2 s before deceleration, 300 μL chlorobenzene antisolvent (preheated to 65 °C) was dripped slowly on the surface of substrate over an interval of 1 s. The substrate was immediately annealed on a hotplate. The annealing process was 45 °C for 10 min, 65 °C for over 10 min, and 100 °C for 10 min.

**Solar cell device fabrication**. Glass/ITO substrates (10 Ω sq$^{-1}$, Geomatec Co., Ltd.) were etched with zinc powder and HCl (6 M in de-ionized water), then consecutively cleaned with water, acetone, detergent solution (Semico Clean 56, Furuuchi chemical), water, and isopropyl alcohol with 15 min ultrasonic bath, followed by drying with an air gun. Finally, the organic residues on substrates were removed with plasma treatment. PEDOT:PSS aqueous dispersion was filtered through a 0.45 μm PTFE filter and then spin coated on the ITO surface at 500 rpm for 10 s and 4000 rpm for 60 s, and then annealed at 140 °C for 20 min. The substrates were transferred to an Ar-filled glove box (O$_2$ < 0.1 ppm) and annealed at 140 °C for another 20 min. The perovskite layer was fabricated on top of PEDOT:PSS following the above-mentioned procedure. For the surface treatment, 0.1 mM solution of EDA in toluene was spin coated at 5000 rpm for 50 s, followed by annealing at 70 °C for 5 min. Subsequently, 1 mg mL$^{-1}$ solution of PC$_{61}$BM in chlorobenzene was spin coated at 5000 rpm for 50 s, followed by annealing at 70 °C for 5 min. Then, 20 nm of C$_{60}$ (0.01 nm s$^{-1}$) and 8 nm of BCP (0.01 nm s$^{-1}$) were deposited by thermal evaporation. Finally, 100 nm of silver (0.005 nm s$^{-1}$) was deposited through a shadow mask to form the metal electrode. The device area was approximately 15 mm$^2$. For the samples for certification, the solar cell devices were sealed with a tesa tape / barrier film (PT7/25GT3) applied at 70 °C for 10 min.

**Characterization**. Photocurrent–voltage ($J$–$V$) curves for PSCs were measured in a nitrogen-filled glovebox (O$_2$ < 1 ppm) with an OTENTO-SUN-P1G solar simulator (BUNKOUKEIKI Co., Ltd.) and a Keithley 2400 sourcemeter. The light intensity of the illumination source was calibrated using a standard BS520 silicon photodiode. The active area of the device was maximized as the device area, and the dimensions of which were measured on a per-device basis. A shadow mask with an area slightly greater than the device was used to protect the other cells on the substrate. EQE and internal quantum efficiency (IQE) spectra were measured by a SMO-250III system equipped with an SM-250 diffuse reflection unit (BUNKOUKEIKI Co., Ltd.). The light intensity of the illumination source was calibrated with a standard SiPD S1337-1010BQ silicon photodiode. XRD measurements were performed on a Rigaku RINT 2500 (Rigaku Co.) diffractometer. Perovskite films were deposited on the surface of PEDOT:PSS with glass/ITO as substrates and covered with a thin film of spin coated poly(methyl methacrylate) (PMMA, Sigma–Aldrich Co.) to prevent direct exposure to air. XPS was recorded with an ESCA-3400HSE (SHI-MADZU Co.) instrument. Scanning electron microscopy (SEM) was performed with an S8010 (Hitachi High-Technologies Co.) instrument. TEM was performed with a JEM-1011 (JEOL Co., Ltd.) instrument. Photoelectron yield spectroscopy (PYS) measurements were carried out using a BUNKOUKEIKI BIP-KV201 (accuracy: ±0.02 eV, extraction voltage = 10 V) under vacuum (<10$^{-2}$ Pa). Perovskite film samples for PYS measurements were prepared by deposition of the precursor solution on the surface of PEDOT:PSS with ITO as substrates in an Ar-filled glove box and transferred to the chamber for PYS measurement without exposure to air. For the time-resolved photoluminescence (TRPL) measurements, the samples were excited by a picosecond pulsed light with a wavelength of 688 nm (Advanced Laser Diode System). The excitation fluence was set at 127 nJ cm$^{-2}$. The PL signals were recorded using an avalanche photodiode (ID Quantique) and a time-correlated single photon counting board (Becker and Hickl). The PL lifetimes were obtained by fitting the PL decay curve with an exponential function. The PL spectra were recorded using an InGaAs array detector equipped with a monochromator (Princeton Instruments). The samples were kept in an Ar-filled metallic box for the whole process to avoid oxygen contamination and degradation. The $^1$H

and [119]Sn NMR measurements were carried out with JEOL JNM-ECA 500 and Bruker Avance III 600US Plus NMR instruments. The chemical shifts were reported in ppm using residual proton signals in the deuterated solvents and using tetramethyltin as a reference for the [1]H and [119]Sn NMR measurements, respectively.

**Reporting summary**. Further information on research design is available in the Nature Research Reporting Summary linked to this article.

## Data availability

The data that support the findings of this study are available from the corresponding author upon reasonable request.

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

## Acknowledgements

This work was partially supported by JST–ALCA (JPMJAL 1603), JST–COI (JPMJCE 1307), JST–CREST (JPMJCR16N3) programs, NEDO, International Collaborative Research Program of ICR, Kyoto University, Grant-in-Aid for Research Activity Start-up (19K23631), Grant-in-Aid for JSPS Fellows (18F18342), and Grant-in-Aid for Scientific Research (C) (19K05666). M.A.T. thanks the JSPS for Postdoctoral Fellowships for Research in Japan. [119]Sn NMR measurements were supported by the Joint Usage/ Research Center (JURC) at ICR, Kyoto University. We thank Prof. Yuichi Shimakawa and Dr. Masato Goto (Kyoto University) for XRD measurement, Prof. Yoshitaro Nose and Dr. Ryoji Katsube (Kyoto University) for XPS measurement, and Prof. Toshiharu Teranishi and Dr. Ryota Sato (Kyoto University) for TEM observation. We thank Dr. Hidenori Saito, Mr. Daisuke Aoki, and Mr. Tomoyuki Tobe (Kanagawa Institute of Industrial Science and Technology; KISTEC) for the measurement for device certification. We also thank Prof. Akinori Saeki (Osaka University) and Prof. Kazuhiro Marumoto (Tsukuba University) for fruitful discussions.

## Author contributions

T.N., S.Y., T.S., and A.W. conceived the idea. S.Y., J.L., and K.O. carried out the PL measurements with help of T.H. and Y.K. T.N. conducted the XPS measurement. S.Y. and K.O. performed the NMR study. T.N., S.Y., M.A.T., K.K., S.H., K.O., and R.H. fabricated the solar cell devices. H.D.K. and H.O. examined the light intensity and temperature dependence of device performance. T.N., R.M., and A.W. prepared the manuscript. All authors commented on the manuscript. A.W. supervised the project.

## Competing interests

The authors declare no competing interests.
