## [Peer Review File · Nature Communications]

Reviewers' comments:

Reviewer #1 (Remarks to the Author):

This work reports an in-situ reduction in the perovskite precursor solution by the highly selective reaction of 1,4-bis(trimethylsilyl)2,3,5,6-tetramethyl-1,4-dihydropyrazine (TM-DHP) with SnF₂. Perovskite layers fabricated with the purified precursor solution were determined by XPS to be essentially free of Sn⁴⁺ ions, and showed strong photoluminescence with prolonged decay lifetimes. Using this nanoparticle treatment, the power conversion efficiency of tin-based solar cells reached 11.5%, with an open-circuit voltage of 0.76 V. This work is interesting and the work is well organized and interpreted, the reported conversion efficiency is among the highest value in the tin halide perovskite solar cells. However, there are some issues that need further investigations before it can be considered for publication in Nature Communications.

1. The author filtered the SnO nanoparticles through a 0.45 μm PTFE filter as mentioned in the Experimental Part. But in Figure 3c, the particle size is less than 10 nm as observed in TEM images. How could a 0.45 μm PTFE filter do this ultrafine filtration?
2. Since the reported PCE is among the highest value in the tin halide perovskite solar cells, a third party certification would be highly recommend to make the results more solid.
3. The author used PCBM with a higher LUMO level to further improve the Voc of the corresponding device, but a thin layer of C60 is still used. The reviewer would wonder the device voltage output in a configuration in absence of C60 layer.
4. The current output is considerable low given the much larger absorption range of the tin halide perovskite. Would this low current related with the limited perovskite film thickness in the p-i-n structure? What are the diffusion lengths of the charger carrier of the tin perovskite layer after in-situ reduction?

Reviewer #2 (Remarks to the Author):

The article describes use of a reducing agent to create Sn nanoparticles from SnF₂, which the authors claim reduces SnI₄ impurities.

The article is interesting, but requires more clarification before publication.

It is not clear why the reaction has to proceed by reacting with the SnF₂ rather than SnI₄ directly.

Also, is any of the reducing agent remaining in solution?

The authors should show results on 1:1 Sn:Pb compositions also - these make much more efficient starting solar cells.

Can the authors describe in more detail why PCBM gives them such higher Vocs?

Reviewer #3 (Remarks to the Author):

In this manuscript, TM-DHP was introduced in the perovskite precursor solution to selectively reduce

SnF₂ to SnO nanoparticles, which can scavenge Sn⁴⁺ impurities and result in 11.5% PCE for the device. The approach of using TM-DHP brings some novelty and the results are interesting. So the paper can be published in Nature Communication subject to a revision on the following aspects:

1. This article reported device with promising PCE of 11.5% with relatively large device area (15mm²). It is suggested to send the device to an independent institution for certification.
2. Can the authors also test the influence of SnF₂ amount on the Sn⁴⁺ content in the perovskite films?
3. Storage stability is not equal to operational stability under light illumination. It is better to give stability data of the solar cells under continuous illumination at maximum power point.

Point-by-Point Response to the Reviewers' Comments

Reply to Reviewer #1:

This work reports an in-situ reduction in the perovskite precursor solution by the highly selective reaction of 1,4-bis(trimethylsilyl)2,3,5,6-tetramethyl-1,4-dihydropyrazine (TM-DHP) with SnF₂. Perovskite layers fabricated with the purified precursor solution were determined by XPS to be essentially free of Sn⁴⁺ ions, and showed strong photoluminescence with prolonged decay lifetimes. Using this nanoparticle treatment, the power conversion efficiency of tin-based solar cells reached 11.5%, with an open-circuit voltage of 0.76 V. This work is interesting and the work is well organized and interpreted, the reported conversion efficiency is among the highest value in the tin halide perovskite solar cells.

We appreciate the overall positive evaluation from Reviewer #1 on our manuscript.

However, there are some issues that need further investigations before it can be considered for publication in Nature Communications.

Q1-1: The author filtered the Sn⁰ nanoparticles through a 0.45 μm PTFE filter as mentioned in the Experimental Part. But in Figure 3c, the particle size is less than 10 nm as observed in TEM images. How could a 0.45 μm PTFE filter do this untafine filtration?

A1-1: Although the size of particles formed just after the addition of reductant **TM-DHP** is less than 15 nm, these particles quickly aggregate during stirring, eventually forming millimeter-sized nuggets of bulk metal. The Sn⁰ may then be conveniently removed by the 0.45 μm PTFE filter. To clarify this point, we added the TEM and optical microscope images of the aggregates at different stirring times to the supporting information as Figures S13. We also changed the main text (page 7) from,

“These Sn⁰ nanoparticles gradually aggregate in the precursor solution. After stirring for 15 min, Sn⁰ bulk metal was precipitated which can be conveniently removed by filtration.”

to,

“These Sn⁰ nanoparticles gradually aggregate in the precursor solution to form larger particles. The nanoparticles grow to ca. 60 nm after 30 s and form a Sn⁰ bulk metal precipitate of >1 mm after stirring for 15 min (Figure S13). After stirring for another 15 min, metal precipitate was conveniently removed by filtration through a 0.45 μm PTFE filter.”

Q1-2: Since the reported PCE is among the highest value in the tin halide perovskite solar cells, a third party certification would be highly recommend to make the results more solid.

A1-2: Following the reviewer's suggestion, we prepared an encapsulated Sn perovskite device which was sent to a third party institution, the Kanagawa Institute of Industrial Science and Technology, for certification. The certified PCE was 11.2%, as shown in Figure S25. We also added a sentence to the main text (page 11) as follows:

"An encapsulated device was tested by a professional institution, giving a certified PCE of 11.2% (Figure S25)."

Q1-3: The author used PCBM with a higher LUMO level to further improve the Voc of the corresponding device, but a thin layer of C60 is still used. The reviewer would wonder the device voltage output in a configuration in absence of C60 layer.

A1-3: Following the reviewer's suggestion, we fabricated Sn perovskite devices without C₆₀. The PCE dropped to only 2.1% with lower device parameters throughout ($J_{SC} = 16.5 \text{ mA cm}^{-2}$, $V_{OC} = 0.36 \text{ V}$, and $FF = 0.36$). Clearly, the C₆₀ layer plays a vital role in the electron extraction and transport. If, for example, the spin-coated PCBM layer does not completely cover the perovskite layer, a vacuum-deposited C₆₀ layer may be needed to fill in pinholes and prevent recombination. We added $J-V$ curves of the PCBM-only device in Figure S23 and added a sentence (page 10) as follows:

"A C₆₀ layer was necessary in our device structure, as confirmed by the much lower performance in its absence (Figure S23)."

Q1-4: The current output is considerable low given the much larger absorption range of the tin halide perovskite. Would this low current related with the limited perovskite film thickness in the p-i-n structure? What are the diffusion lengths of the charger carrier of the tin perovskite layer after in-situ reduction?

A1-4: We checked the device performance for three different thicknesses of the perovskite layer, 190 nm, 270 nm, and 320 nm. As shown in Figure S19, the short circuit current did not increase as a result, indicating that the carrier diffusion length of the tin perovskite is not more than about 200 nm. We added the above discussion to the manuscript (page 9) as follows:

"Increasing the thickness of the perovskite layer to 270 nm or 320 nm did not result in

any increase in J_{SC} , suggesting that the charge carrier diffusion lengths do not exceed 200 nm (Figure S19)."

Reply to Reviewer #2:

The article describes use of a reducing agent to create Sn nanoparticles from SnF₂, which the authors claim reduces SnI₄ impurities.

The article is interesting, but requires more clarification before publication.

Q2-1: It is not clear why the reaction has to proceed by reacting with the SnF₂ rather than SnI₄ directly.

A2-1: As we have already shown in Figure 3, the NMR results confirm that **TM-DHP** reacts selectively with SnF₂ compared with SnI₄. This is interesting because the reduction potentials would suggest that SnI₄ is preferentially reduced. The selectivity for SnF₂ is likely the result of the strong affinity of the trimethylsilyl groups with fluoride, as discussed on page 7 in the main text.

Q2-2: Also, is any of the reducing agent remaining in solution?

A2-2: Considering the fast reaction of the reductant **TM-DHP** with SnF₂, reductant itself does not remain after stirring for 30 min. In order to confirm this, we conducted an additional NMR measurement on the 1:1 mixture of **TM-DHP** and SnF₂ in DMSO-*d*₆, which was added as Figure S11b. We confirmed that the reductant **TM-DHP** completely reacts with SnF₂ in solution to form tetramethylpyrazine (**TMP**) and SiMe₃F. Considering the fact that the boiling points of the formed **TMP** (bp. 190 °C) and SiMe₃F (bp. 16 °C) are comparable or lower than DMSO (solvent, bp. 189 °C), these would hardly remain in the perovskite layer after annealing process. We also confirmed the fact that the addition of **TMP** to precursor solution did not significantly affect the PL lifetimes, as shown in Figure S7.

Q2-3: The authors should show results on 1:1 Sn:Pb compositions also - these make much more efficient starting solar cells.

A2-3: The focus of our research is achieving lead-free perovskite photovoltaic materials. Of course, adding lead to the tin perovskite precursor solution will result in increased device performance, and we note that our Sn⁴⁺-scavenging method would also be

applicable for the Sn/Pb-mixed perovskite compositions of this type. Although the 1:1 Sn/Pb composition is an interesting target, we think it is out of the scope of this paper.

Q2-4: Can the authors describe in more detail why PCBM gives them such higher Vocs?

A2-4: As shown in the energy level diagram in Figure 5b, there is a large energy difference between the conduction band of perovskite (−3.82 eV) and the LUMO level of C₆₀ (−4.21 eV). The insertion of PCBM, which has higher LUMO energy level (−4.00 eV) than C₆₀, aligns more closely with the conduction band of the tin perovskite, which leads to suppressed charge carrier recombination at the interface and improved V_{OC} as a result. This approach was also demonstrated by the recent paper (Ning, Z. et al., *Nat. Commun.* **2020**, added as ref. 34 in the main text) which was published while our manuscript was being revised. To make this point clear, we modified the sentence in page 10 from, *“Furthermore, a thin layer of PC₆₁BM (< 5 nm) was inserted between the perovskite and the C₆₀ layers to improve the energy level matching (Figure 5b, S21), and to reduce non-radiative recombination at the interface.”*

to,

“Furthermore, a thin layer of PC₆₁BM (< 5 nm) was inserted to promote Ohmic contacts by minimizing the energy gap between the conduction band of perovskite and the LUMO energy level of the electron-transporting layer (Figure 5b, S21), and to reduce non-radiative recombination at the interface.³⁴”

Reply to Reviewer #3:

In this manuscript, TM-DHP was introduced in the perovskite precursor solution to selectively reduce SnF₂ to SnO nanoparticles, which can scavenge Sn⁴⁺ impurities and result in 11.5% PCE for the device. The approach of using TM-DHP brings some novelty and the results are interesting. So the paper can be published in Nature Communication subject to a revision on the following aspects:

We appreciate the overall positive evaluation from Reviewer #3 on our manuscript.

Q3-1: This article reported device with promising PCE of 11.5% with relatively large device area (15mm²). It is suggested to send the device to an independent institution for certification.

A3-1: According to the suggestion of reviewers 1 and 3, we prepared an encapsulated Sn

perovskite device which was sent to a third party institution, the Kanagawa Institute of Industrial Science and Technology, for certification. The certified PCE was 11.2%, as shown in Figure S25. We also added a sentence to the main text (page 10) as follows:

“An encapsulated device was tested by a professional institution, giving a certified PCE of 11.2% (Figure S25).”

Q3-2: Can the authors also test the influence of SnF₂ amount on the Sn⁴⁺ content in the perovskite films?

A3-2: We thank the reviewer for this valuable suggestion. To test this, we prepared perovskite films from precursor solutions containing 10, 5, and 1 mol% SnF₂, to which 1 mol% **TM-DHP** was added. As shown in Figure S10, XPS measurements reveal that the bulk perovskite is free of Sn⁴⁺, confirming the effectiveness of our Sn⁴⁺ scavenging method even with 1 mol% SnF₂. However, Sn⁴⁺ content at the perovskite layer surface decreased as the SnF₂ amount was increased. This shows the positive effect of excess SnF₂ to suppress surface oxidation occurring after fabrication.

This discussion was included in the manuscript (page 5) as follows:

*“The effect of varying the amount of SnF₂ was also investigated. The bulk perovskite remained Sn⁴⁺-free even when the amount of SnF₂ was reduced to 1 mol% (Figure S10). The Sn⁴⁺ content at the perovskite surface decreased when the SnF₂ amount was increased, however, showing the positive effect of excess SnF₂ to suppress surface oxidation occurring after film fabrication. Optimal results were obtained with the ratio of 10 mol% SnF₂ and 1 mol% **TM-DHP**.”*

Q3-3: Storage stability is not equal to operational stability under light illumination. It is better to give stability data of the solar cells under continuous illumination at maximum power point.

A3-3: According to reviewer's suggestion, the stability data for 600 s for a representative Sn-based perovskite solar cell under continuous illumination at a bias of 0.49 V was added in Figure S17, confirming the reliability of the device parameters obtained from the *J–V* curves. We also added a sentence to the manuscript (page 9) as follows:

“Stable power output for 600 s under AM1.5G operation was confirmed (Figure S17).”

In addition to the above response to the reviewers, we modified the manuscript as follows:

- Ruito Hashimoto was added as a coauthor for the contribution to the additional experimental work.
- We also added some details to the experimental section and made some additional acknowledgements.

Atsushi Wakamiya
Kyoto University

REVIEWERS' COMMENTS:

Reviewer #1 (Remarks to the Author):

The authors addressed all the comments from the reviewer and it can be accepted for publication now.

Reviewer #2 (Remarks to the Author):

The authors have addressed my comments fully and the article should be published with no further revisions necessary.

Reviewer #3 (Remarks to the Author):

The revised version is OK for publication.